# Citric Acid Promotes Immune Function by Modulating the Intestinal Barrier

**DOI:** 10.3390/ijms25021239

**Published:** 2024-01-19

**Authors:** Pengcheng Hu, Meng Yuan, Bolun Guo, Jiaqi Lin, Shihong Yan, Huiqing Huang, Ji-Long Chen, Song Wang, Yanmei Ma

**Affiliations:** 1Joint Laboratory of Animal Pathogen Prevention and Control of Fujian-Nepal, College of Animal Sciences, Fujian Agriculture and Forestry University, Fuzhou 350002, China; hupengcheng@fafu.edu.cn (P.H.); yuanmeng.97@foxmail.com (M.Y.); shihongyan0106@163.com (S.Y.); chenjilong@fafu.edu.cn (J.-L.C.); wangsong@fafu.edu.cn (S.W.); 2Key Laboratory of Fujian-Taiwan Animal Pathogen Biology, College of Animal Sciences, Fujian Agriculture and Forestry University, Fuzhou 350002, China; guobolun@fafu.edu.cn (B.G.); linjiaqi@fafu.edu.cn (J.L.); huiqing447@163.com (H.H.)

**Keywords:** citric acid, intestinal barrier, H9N2 influenza virus, gut microorganisms, metabolomics

## Abstract

Amidst increasing concern about antibiotic resistance resulting from the overuse of antibiotics, there is a growing interest in exploring alternative agents. One such agent is citric acid, an organic compound commonly used for various applications. Our research findings indicate that the inclusion of citric acid can have several beneficial effects on the tight junctions found in the mouse intestine. Firstly, the study suggests that citric acid may contribute to weight gain by stimulating the growth of intestinal epithelial cells (IE-6). Citric acid enhances the small intestinal villus–crypt ratio in mice, thereby promoting intestinal structural morphology. Additionally, citric acid has been found to increase the population of beneficial intestinal microorganisms, including Bifidobacterium and Lactobacillus. It also promotes the expression of important protein genes such as occludin, ZO-1, and claudin-1, which play crucial roles in maintaining the integrity of the tight junction barrier in the intestines. Furthermore, in infected IEC-6 cells with H9N2 avian influenza virus, citric acid augmented the expression of genes closely associated with the influenza virus infection. Moreover, it reduces the inflammatory response caused by the viral infection and thwarted influenza virus replication. These findings suggest that citric acid fortifies the intestinal tight junction barrier, inhibits the replication of influenza viruses targeting the intestinal tract, and boosts intestinal immune function.

## 1. Introduction

Due to the overuse of antibiotics, there is a growing concern regarding increasing antibiotic resistance [1]. Therefore, replacements for antibiotics as alternatives are attracting attention [2,3]. One of the alternative replacements is citric acid (CA, C6H8O7), an organic acid produced naturally in the body through the synthesis of acetyl coenzyme A and oxaloacetate. CA plays a crucial role in the metabolism of sugars, fats, and amino acids [4]. A study conducted by Suzuki explored the use of a high-pressure gaseous mixture (HPG) consisting of carbon monoxide and oxygen. The mixture was found to reduce oxidative stress, resulting in cardiac protection by enhancing aerobic metabolism in vivo. Notably, the study revealed a significant increase in CA levels in the HPG group [5]. CA as a supplement in animal feed has a rich history dating back to 1937 when Pileggi et al. discovered its positive effects on rat health, improving phosphorus utilization and preventing achondroplasia [6]. Further research by Li demonstrated the diverse inhibitory and inducing effects of CA on intestinal microorganisms [7]. Additionally, a feed additive blend containing CA has been shown to reduce levels of aerobic bacteria, fungi, and *E. coli* in the small intestine of broiler chickens. This blend also enhances the length of intestinal villi and the depth of crypts [8]. In broilers suffering from necrotizing enteritis, the introduction of CA into their diet contributed to a decrease in pathogenic bacteria within the intestinal tract, an improvement in the intestinal microbiota, and a reduction in the inflammatory response [9]. Moreover, Jeong et al. demonstrated that CA possesses inhibitory effects on the secretion of the pro-inflammatory factors TNF-α and interleukin IL-6 by LPS/IFN-γ stimulated macrophages [10].

The intestine comprises intestinal-associated lymphoid tissue, which serves as a crucial defense mechanism in animals, protecting against harmful substances and pathogens. Evaluating the effectiveness of the intestinal barriers is essential for assessing intestinal health. The intestinal barrier comprises physical, chemical, microbial, and immune components that work together to fend off invaders [11]. The physical barrier is composed primarily of intestinal epithelial cells and their tight junctions. These junctions are situated at the outer membrane side of the intestinal epithelial cells, with the aim of closing the intercellular gap, which helps prevent toxic substances from entering the surrounding tissues. Additionally, they also regulate the transport of substances between the cells and inhibit the spread of pathogens [12,13]. Tight junctions are formed by approximately 50 proteins including occludin, claudins, and zonula occludens (ZO). These junctions exhibit a dynamic structure that can adjust the tightness in response to various external stimuli and physiological and pathological responses. Maintaining the integrity of these tight junctions between the intestinal mucosal villous cells is crucial for preserving the function of the intestinal barrier. Apart from the physical barrier, a significant amount of symbiotic flora in the intestinal tract adheres to the intestinal mucosa, forming a microbial barrier. The normal flora maintains a mutually beneficial relationship with the host—providing nutrients for host digestion and preserving the balance of the intestinal ecosystem. Certain anaerobic bacteria, like Lactobacillus and Bifidobacterium, tightly adhere to the intestinal epithelium, competitively inhibiting the binding of pathogenic bacteria to the intestinal epithelium [14]. Intestinal microorganisms secrete metabolites, such as acetic acid, lactic acid, and butyric acid, which lower the pH of the intestine, inhibiting the growth of pathogenic bacteria. Metabolites produced by *Bifidobacterium bifidum* also inhibit the production of tumor necrosis factor by the intestinal epithelial cells. Notably, butyric acid promotes the proliferation of intestinal epithelial cells [15]. Maintaining a healthy microbial barrier, promoting proper intestinal growth, and ensuring the integrity of the mechanical barrier are crucial for overall organismal health.

Influenza viruses mainly inhabit the lungs and cause respiratory symptoms, frequently accompanied by some concurrent intestinal symptoms [16]. Viruses can also be found in fecal samples [17,18]. The H9N2 subtype of avian influenza virus (H9N2 AIV) is a virus of low pathogenicity [19]. Following influenza virus infection, Escherichia coli commonly causes bacterial secondary infection, leading to disturbances in the intestinal flora, resulting in structural changes and inflammation [20].

Metabolomics is an emerging method in histology, gaining popularity in veterinary medicine. Researchers use this approach to identify metabolic changes in organisms and assess the impact of diet on metabolism [21]. Studies have revealed an association between gut microbiota and serum metabolism [22], with the changes in serum metabolites having varied effects on the organism [23]. CA is frequently used as a growth enhancer in animal and poultry feeds, yielding significant results. However, the effect of CA on influenza virus infection of the intestinal lining is still elusive.

To investigate the implication of CA on the intestinal lining during the influenza virus, this study utilized C57BL/6 mice and mouse intestinal epithelial cells (IEC-6) for experimental observation. A comprehensive experiment, including RT-PCR, RT-qPCR, flow cytometry, and metabolomics analysis, was employed to explore the molecular impact of CA on the organisms, aiming to establish a scientific foundation for the application of CA either alone or in combination with other additives.

## 2. Results

### 2.1. Citric Acid Increases Body Weight and Stimulates Cell Growth by Increasing the Number of G2 Phase Cells in IEC-6 Cells of Mice

To examine the impact of citric acid on the development of mice and intestinal cells, diverse concentrations of citric acid were administered to both mice and intestinal epithelial cells, followed by cultivation. The findings indicate that Figure 1 illustrates the result obtained from monitoring the body weight and cell growth cycle of mice in different groups. In Figure 1A, it can be observed that the body weight of the mice increased progressively over the duration of the experiment in all groups. Compared to the control group (0 g/L), the body weights of the mice in both groups with concentration levels of 0.05 g/L and 0.1 g/L increased. Notably, the group with a concentration level of 0.1 g/L exhibited the most significant weight gain among the test groups. Figure 1B presents the cell growth data, indicating no significant difference in cell number between the test groups during the initial 12 h incubation period. However, after 24 h of cell culture, all test groups showed an increased cell number compared to the control group (0 g/L). In Figure 1C, there is a significant increase in cell numbers in the concentration groups of 0.1 g/L, 0.15 g/L, and 0.2 g/L after 24 h, in comparison to the control group.

Figure 1D displays flow cytometry plots illustrating different cell cycle groups, with abbreviations explained upon first use. The G1 phase contains the majority of cells represented by a peak, while a smaller percentage of cells are in the G2 phase. The S phase lies between the two peaks, forming a blunt, rounded peak.

Comparison of the cell cycle in Figure 1E,F reveals a significant decrease in the count of cells in the G1 phase of the experimental group and a corresponding increase in the count of cells in the G2 phase when compared to the control group (0 g/L). Notably, a highly significant increase was observed in the concentration group of 0.15 g/L, while a significant increase was observed in the concentration groups of 0.1 g/L and 0.2 g/L.

### 2.2. Improvement of Intestinal Structure by Citric Acid-Induced Increase in the Ratio of Villi to Crypts in Mouse Small Intestine

The gut’s physical barrier is mainly composed of epithelial cells and their tight junctions. Our investigation focused on CA’s effects on the structure and morphology of the gut. Figure 2 illustrates the HE-stained sections of the small intestine, showing the presence of visible, intact small intestinal villi in all groups of mice free from lesions, with clear layers of villous tissue and undamaged epithelial cell boundaries. Table 1 demonstrates the data on the crypt ratio of the intestinal villi, indicating a significant increase in all experimental groups compared to the control group (0 g/L). A highly significant increase was seen in the 0.1 g/L and 0.15 g/L concentration groups.

### 2.3. Citric Acid Enhances Growth of Common Probiotic Bacteria in the Cecum of Mice

The maintenance of the intestinal barrier is heavily reliant on the stability of the intestinal bacterial flora. Hence, a study was undertaken to investigate the impact of CA on the intestinal flora. The contents of the mice’s cecum were analyzed following the administration of CA. The bacterial quantification results for each group are presented in Table 2. In comparison to the control group (0 g/L), the concentration of Bifidobacteria in the group with a 0.1 g/L concentration exhibited a remarkably significant increase. Moreover, the concentration of Lactobacillus showed an upward trend in the groups with concentrations of 0.1 g/L, 0.15 g/L, and 0.2 g/L, although the difference was not statistically significant. The analysis of the Escherichia coli revealed no significant difference in the number of bacteria within the various groups.

### 2.4. Citric Acid Induced the Expression of Tight Junction Genes in Intestinal Tissue and IEC-6 Cells

The outcomes of our former experiments have demonstrated that CA has the ability to promote cellular growth and enhance the morphological and structural alterations of the intestine. In order to delve deeper into the impact of CA on tight junctions, variations in the expression of the tight junction genes were monitored. Figure 3 shows the mRNA expression of the tight junction genes occludin, claudin-1, and ZO-1 in both intestinal tissues and IEC-6 cells from mice administered with CA. Figure 3A–D shows that in the intestinal tissues of mice, the mRNA expression of occludin, ZO-1, and claudin-1 was significantly or highly significantly increased in the 0.05 and 0.1 g/L concentration groups compared with the control group (0 g/L). Figure 3E–H showed that in IEC-6 cells, occludin mRNA was highly significantly elevated in all concentration groups compared with the control group (0 g/L), with the most notable increase observed in the 0.1 g/L concentration group. Furthermore, the mRNA expression of ZO-1 and claudin-1 mRNA showed a significant increase in the 0.1 g/L concentration group in Figure 3G,H.

### 2.5. Transcriptome and Metabolic Pathway Analysis of Citric Acid-Induced Serum Metabolism in Mice

To explore the impact of citric acid on the body’s metabolism, we obtained serum samples from mice fed with CA for serum metabolomics analysis in this investigation. Figure 4 presents the results of SIMCA analysis. Findings from the PCA and OPLS-DA analyses (Figure 4A,B) indicate no significant overlap and no substantial differences between the two sample groups, indicating reliable and significant differences in the data obtained from these groups. A 200 permutation test was performed on the data shown in Figure 4C, with the left side displaying lower Q2 and R2 values compared to the right side, indicating a valid model validation. Therefore, the result can be considered reliable. Differential metabolites were selected by S-plot analyses if their VIP score exceeded 1. Figure 4D indicates that a total of 79 such metabolites were detected. A statistical analysis screened 60 significant differential metabolites (*p* < 0.05), comprising 35 significantly up-regulated and 25 significantly down-regulated metabolites. The metabolites were clustered and subjected to analysis, and the clustering heat map is displayed in Figure 5A. MetaboAnalyst 5.0 (MetPA) was then used for the analysis of differential metabolites. Figure 5B and Table 3 display the results and enrichment profiles of the MetPA metabolic pathway analysis. The analysis showed that a total of 12 pathways were involved, and the pathways with an effect score greater than 0.1 were selected for analysis [24]: taurine metabolism, arachidonic acid metabolism, and amino acid metabolism. Figure 5C–E exhibits the KEGG pathway diagrams of the metabolic pathways that were evaluated for significance.

### 2.6. Citric Acid Maintains Tight Junction Gene Expression in IEC-6 Cells during H9N2 Infection and Inhibits Viral Replication

Numerous studies have demonstrated that influenza viruses can be detected within the intestinal tract. To examine the impact of H9N2 on intestinal epithelial cell production, we infected IEC-6 with H9N2. Figure 6 illustrates that the expression of inflammatory factors TNF-α and interleukin IL-6 was significantly or very significantly elevated within 24 h after IEC-6 cells were infected with H9N2. In addition, the expression of type I interferon IFN-β and its downstream gene ISG15 also showed similarly elevated expression. The study indicates that H9N2 induces the production of inflammatory agents and interferes with tight junction protein genes in cells. Subsequently, the research delved into the capacity of CA to protect against H9N2 infection in IEC-6-treated cells. Figure 7A–C shows a significant decrease in the expression of the intracellular tight junction gene claudin-1 upon infection with H9N2 AIV. Furthermore, there is a significant increase in the expression of all three tight junction genes in the experimental CA-added group. The expression of the inflammatory factors TNF-α and interleukin IL-6 in H9N2-infected IEC-6 cells was significantly reduced by the addition of citric acid, and viral proliferation was inhibited by a significant increase in type I interferon IFN-β and the interferon-stimulated gene ISG15 (Figure 7D–H).

## 3. Discussion

The experiment demonstrates the growth-promoting effect of CA. The addition of 0.05 and 0.1 g/L CA resulted in increased body weight of mice compared to the control group (0 g/L), with 0.1 g/L CA demonstrating the strongest effect. The observed phenomenon may be ascribed to the acidic pH levels and suboptimal palatability, which ostensibly diminished the water intake and appetite in mice, thereby leading to a decelerated increase in body mass. Body mass gain is affected by various factors, including feed intake, changes in temperature, and genetic factors [25,26]. The growth curves of the cells demonstrated that the control group (0 g/L) cells reached a logarithmic growth phase in 24–36 h, while the cells added with 0.1 and 0.15 g/L citric acid entered this phase in 12 h. At 24 h, there was a significant increase in the cell count for the concentration groups of 0.1 g/L, 0.15 g/L, and 0.2 g/L compared to the control group (0 g/L). The findings from the 24 h cell cycle assay demonstrated a significant decrease in the percentage of cells in the G1 phase, and a significant increase in cells in the G2 phase in the group where CA was added, in comparison to the control group (0 g/L). The concentration groups of 0.1, 0.15, and 0.2 g/L showed a significant or very significant increase. The cell cycle assay results were in line with those of the 24 h cell growth study, suggesting that CA doses of 0.1, 0.15, and 0.2 g/L can stimulate IEC-6 cell proliferation by accelerating logarithmic phase proliferation and increasing the G2 phase cell percentage, much like the total saponins action principle [27]. The research demonstrates that citric acid acts as a stimulant for the proliferation of epithelial cells, which subsequently strengthens the integrity of tight junctions between intestinal epithelial cells and associated tight junction proteins.

The longer the intestinal villi in the intestine, the shallower the crypts, and the greater the ratio of villi to crypts, the greater the digestive and absorptive capacity of the small intestine [28,29]. In the conducted experiment, all test groups with added citric acid exhibited an increase in the chorionic crypt ratio to some extent. Notably, the concentration groups with 0.1 g/L and 0.15 g/L had significantly increased ratios. The results imply that incorporating 0.1 g/L and 0.15 g/L concentrations of citric acid may elevate the villus-to-crypt ratio in the murine intestine, thereby augmenting digestive and absorptive functions and ameliorating the structural configuration of the intestinal tract. Consistent with previous research [8,30,31,32], these findings propose that various organic acids, such as CA, have the potential to impact the ratio of intestinal villous-to-crypt, which in turn has a positive effect on digestion and absorption, facilitates the growth of intestinal epithelial cells, and ensures the stability of intestinal tight junctions.

The gut microbiota plays a crucial role in digestion, absorption, and metabolism. The combination of CA acidifiers with plant polysaccharides and antimicrobial peptides led to significant changes in the intestinal microbial diversity of broilers, leading to an increased abundance of microorganisms in the cecum [33]. The enhanced abundance of intestinal flora has been found to enhance growth performance in weaned piglets [34]. Additionally, supplementation of preterm infants with Bifidobacterium bifidum significantly increases the abundance of fecal flora and promotes weight gain [35]. The research observed that supplementing with 0.1 and 0.15 g/L concentrations of CA led to a pronounced proliferation of intestinal bifidobacteria in comparison to the control group (0 g/L). Several studies have shown that changes in gut microbiology are important in affecting the gut barrier. The addition of inactivated Lactobacillus to broilers resulted in a notable enhancement of their intestinal barrier [36]. This indicates that citric acid may have promoted gut microbes and therefore improved the tight junction barrier in our experiments.

Intestinal epithelial cells act as a mechanical barrier in the intestinal mucosa. The experiment showed that the addition of CA significantly increased the expression of occludin, claudin-1, and ZO-1 proteins in both mouse small intestinal tissue and IEC-6 cells, suggesting an improvement in IEC-intestinal tight junction barrier function. Consistent with previous research, mixed feed acidifiers improved the tight junction barrier of cells. A combination of 1 g/L CA, sorbic acid, thymol, and vanillin improved tight junction barrier function in cco-2 cells [37]; and CA significantly or markedly increased ileal mRNA expression of occludin, claudin-1, and ZO-1 in piglets [38,39].

Measurement of serum metabolites by CA revealed that administering 0.1 g/L CA caused alterations in 12 metabolic pathways of serum metabolism in mice. Taurine and hypotaurine metabolism, arachidonic acid metabolism, and cysteine and methionine metabolism were the metabolic pathways with pathway effect values exceeding 0.1. Metabolites involved in the above pathway are taurine (C00245), arachidonic acid (C00219), 11,12-epoxydicarbotrienoic acid (11,12-eet, C14770), and l -methionine (C0073). Taurine and hypotaurine metabolism have the greatest influence among the serum metabolites affected by CA. The metabolite implicated in these biochemical processes is taurine, a sulfur-containing amino acid that is critical for human growth and developmental processes [40]. It has been demonstrated that taurine and other metabolites formed in vivo produce compounds that inhibit IκB kinase activity and NF-κB activation. Taurine chloramine (Tau-Cl) has been found to attenuate IKK kinase activity and reduce the phosphorylation of IKK protein complexes by signaling upstream of the Ikk kinase. This causes targets to undergo ubiquitination and protease-mediated degradation, thereby inhibiting the activation of NF-κB translocation to the nucleus [41]. The process additionally triggers the generation of heme oxygenase-1 (HO-1), an antioxidant enzyme coordinated by Nrf2-ARE that safeguards cells against oxidative stress [42]. One study suggests that arachidonic acid improves insulin sensitivity and prevents the onset of diabetes in rats fed a high-fat diet [43]. Among the metabolites of the CYP pathway, 12-EET has been discovered to enhance the survival rate of mice suffering from heart failure; it inhibits pulmonary fibrosis through the modulation of pro-fibrotic signaling [44]. Additionally, it exerts a diastolic effect and promotes the generation of neo-tissue [45,46]. The experiment revealed the involvement of arachidonic acid and 11,12-epoxy dicarbotrienoic acid in metabolism. It is postulated that CA enhances the metabolic pathway of arachidonic acid, specifically cytochrome p450-11,12-eet, promoting tissue regeneration and boosting the body’s ability to fight against diseases.

A model was constructed to investigate the function of citric acid in the maintenance of the intestinal barrier amidst influenza virus infection. The outcomes of the study indicate that the infection of IEC-6 cells with H9N2 avian influenza virus subtypes precipitated an upregulation of pro-inflammatory cytokines, specifically tumor necrosis factor-alpha (TNF-α) and interleukin-6 (IL-6), along with elevated expression levels of type I interferon, IFN-β, and the interferon-stimulated gene, ISG15. The incorporation of CA facilitates the expression of tight junction genes in H9N2-infected IEC-6 reducing cell permeability, upholding intercellular tight junctions, and conserving the integrity of the intestinal barrier. Hiroko et al. [47] demonstrated the modulation of paracrine transport potential in monolayer cells by IFN, ultimately affecting cell permeability. Additionally, CA reduced the expression of inflammatory cytokines TNF-α and IL-6 in H9N2-infected cells, thus mitigating the inflammatory response to H9N2 IAV infection in an IEC-6 cell. Among various inflammatory responses resulting from stimuli, it was demonstrated by Xilan Tang et al. that citric acid and L-malic acid reduced TNF-α activity and apoptosis via the P13K/Akt pathway [48]. Abdel-Salam demonstrated that CA diminishes the oxidative stress provoked by LPS in the brain and liver of mice, leading to a reduction in TNF-α and paraoxonase (PON1) activity [49]. Zhao et al. have demonstrated that CA can suppress TLR gene expression and alleviate intestinal inflammation by exerting anti-inflammatory effects and preserving intestinal junction tightness via the activation of TLR-mediated MyD88, IRF-3, and NFκB signaling pathways [50]. The results of the study indicate that CA increased the expression of IFN-β and ISG15 in IEC-6 infected with H9N2 and hindered the replication of the influenza virus infection. It has been demonstrated that CA has the capability to improve antioxidant capacity by enhancing the Nrf-2/HO1 pathway [51]. Additionally, 5-Aminolevulinic acid (5-ALA) and sodium ferrous citrate (5-ALA/SFC) induce HO-1 to reduce the cytokine storm induced by SARS-CoV-2 infection to a certain extent [52]. Further studies have shown that HO-1 activators can promote the interaction of HO-1 with IRF3 to induce IFNα/β production to inhibit influenza A virus infection [53]. In another study, it was found that in cells with a weak type I interferon response, the inhibitory effect of HO-1 on IAV may be its mediated oxidative stress response [54]. In conclusion, CA protects the intestinal barrier by enhancing tight junctions and reducing inflammation while also inhibiting viral replication by increasing the antioxidant capacity of HO-1 to induce the production of IFN-β.

## 4. Materials and Methods

### 4.1. Cells, Animals and Reagents

Mouse intestinal epithelial cells (IEC-6) were sourced/obtained from the Cell Resource Center at the Institute of Basic Medical Sciences of the Chinese Academy of Medical Sciences. The 6–8 week-old SPF C57BL/6 male mice, maintained under specific pathogen-free conditions, and feed were obtained from Wu’s Laboratory Animal Trading Co., Ltd. in Minhou County, China. The reverse transcription kit was purchased from Nanjing Novozymes Biotechnology Co. (Nanjing, China). The gum recovery kit, fecal genomic DNA extraction kit, and DEPC water were acquired from Beijing Tiangen Biochemical Technology Co. (Beijing, China). Primers were developed/designed based on the gene sequences published by GenBank (refer to Table 4) and synthesized by Fuzhou Dynco Biotechnology Co. (Fuzhou, China).

### 4.2. Grouping of Mice and Plotting of Body Weight Curves

Thirty SPF mice with similar body weight were randomly assigned to five groups: one control group (0 g/L) and four experimental groups, with six mice in each group. The mice were provided unrestricted access to a diet consisting of 130 high-pressure feeds. The test groups of mice were given varied concentrations of CA solutions in drinking water (0.05 g/L, 0.1 g/L, 0.15 g/L, and 0.2 g/L), while the control group (0 g/L) was given clean drinking water. The mice were placed in the EVC feeding system and reared for a routine period of 14 days. At the end of 14 days, various indices were measured in the mice. The entire experiment was repeated thrice. At the beginning of the experiment, the weight of all sets of mice was noted, and the mean weight was computed as their initial weight. Subsequently, each set of mice was weighed at dawn on the third, sixth, eighth, twelfth, and fourteenth days, and the mean weight of every set was computed and documented.

### 4.3. Cell Assay

#### 4.3.1. Cell Aggregation and Growth Curve Graphing

IEC-6 cells were cultured under normal conditions in a 96 well cell culture plate, with one control group (0 g/L) and four test groups. The control group (0 g/L) was cultured without any additional substances, while the test groups were treated with different concentrations of CA (0.05 g/L, 0.1 g/L, 0.15 g/L, and 0.2 g/L). The cell proliferation was detected using CCK8 assay at 12 h intervals. The optical density (OD) values at each time point were recorded at each time point to generate a cell growth curve.

#### 4.3.2. Cell Cycle Assay

The cells were cultured for 24 h and then treated with 0.25% trypsin. Post digestion, the cells were released through 2 mL of PBS in an ice bath and collected into a sterile centrifuge tube. The tube was centrifuged at 8000× *g* for 3 min, and the supernatant was discarded. The cell pellet was resuspended with 1 mL of ice-cold PBS. The supernatant was then removed again, and the cell precipitates were resuspended using 1 mL of 70% ethanol. The resulting mixture was stored at −20 °C and left overnight in a refrigerator at 4 °C. The stationary cell suspension was subjected to centrifugation at 1000× *g* for 3 min after which the supernatant was aspirated with caution. Single cells were then resuspended by supplementing 1 mL of PBS stored in an ice bath. Thereafter, the cells were centrifuged again to eliminate any remaining supernatant and resuspended again in a propidium iodide staining solution following instructions provided. The fixed and stained cells were analyzed for cell cycle changes using a Beckman Coulter Ultra-High Speed Flow Cell Sorting System (Beckman Coulter Life Sciences, Indianapolis, IN, USA).

### 4.4. Detection of Morphological Structure of Mouse Intestine

On day 14, mice were euthanized and 1 cm of their intestinal tubes were extracted, specifically 10 cm from the pylorus of the stomach. The extracted tubes were washed with 1× PBS solution and immersed in tissue fixative for 24 h. After fixation, the intestinal tubes were sent to Wuhan Sevier Company (Wuhan, China) for preparation of HE tissue section and scanning. CaseViewer software version 2.4 was used to measure the length of the villi and the depth of the crypts of each sample. The villi-to-crypt ratio was calculated accordingly.

### 4.5. Detection of Bacterial Flora in the Contents of Mouse Cecum

The entire cecum of mice was extracted, and then its contents were collected into sterile EP tubes. The genomic DNA of the contents was extracted using a fecal genomic DNA extraction kit. The resulting fecal genomic DNA sample was used as a template for amplifying PCR of 16S rRNA gene fragments of three species by employing primers for Bifidobacterium, Lactobacillus, and Escherichia coli, as per Table 4. The amplified bands were recovered from the target genes using a gel recovery kit. Subsequently, the standards were obtained and their copy numbers were subsequently calculated. Copy number calculation was performed using the following formula:

Copy number = Avogadro constant × fragment concentration/fragment molecular weight.
Copy number (copies·μL^−1^) = (6.02 × 10^23^ × (ng·μL^−1^ × 10^−9^)/(DNA length × 660)).



The RT-qPCR analysis was conducted utilizing a LightCycler96 instrument (Roche, Shanghai, China) which produced a standard curve with copy number plotted on the *x*-axis and CT value on the *y*-axis. Using the standard curve, DNA samples extracted from the cecum contents underwent RT-qPCR quantification, and each bacterial group’s content in the samples was determined based on their Cq values.

### 4.6. Intestinal Tight Junction Assay

#### 4.6.1. Detection of Mouse Intestinal Tight Junction Genes

RNA was extracted from the ileal section of the small intestine and then reverse transcribed. Following that, RT-PCR and RT-qPCR were performed, using the cDNA samples as templates and the occludin, ZO-1, and claudin-1 sequences as primers, which are specified in Table 4. The RT-PCR products were subsequently analyzed via agarose gel electrophoresis, and the target bands were detected using a gel imager. The results from RT-qPCR were analyzed only after the completion of the study’s RT-qPCR results.

#### 4.6.2. Cellular Tight Junction Protein Gene Expression Assay

After 24 h of cell growth, NucleoZol was added, the cells were aspirated, and the lysate was collected. The extracted cellular RNA was reverse transcribed, and RT-PCR and RT-qPCR were performed to detect the expression of three closely related genes, occludin, ZO-1, and claudin-1; the primer sequences are shown in Table 4.

### 4.7. Detection of Serum Metabolites in Mice Using Untargeted Metabolomics

Four mice were taken from each of the control group (0 g/L) and 0.1 g/L CA groups, whole blood was collected and centrifuged at 8000× *g* for 5 min after 30 min at 37 °C, and the serum was collected and stored in a refrigerator at −80 °C overnight and sent to Wuhan Seville Biotechnology Company for metabolite detection by Q-Orbitrap high-resolution liquid chromatography-mass spectrometry. Metabolites with a mzCloud database score of 80 or more were selected and analyzed using SIMCA-P14.1 software. The significance of the samples was tested using *t*-test and one-way ANOVA, and the significant metabolites with *p* < 0.05 and VIP > 1 were selected for metabolite pathway analysis using MetaboAnaiyst version 5.0 software (https://www.metaboanalyzer.ca/, accessed on 15 April 2022).

### 4.8. Detection of Inflammatory Factors and Type I Interferon in IAV-Infected IEC-6 Cells

It was divided into two test groups: blank group and two test groups of no citric acid and added citric acid in the medium. The appropriate amount of cells was evenly distributed in 6 well plates 24 h in advance. The cells were infected with influenza virus when the cells grew to the appropriate density, NucleoZol was used at different time points, and the cells were blown down and the lysates were collected. The extracted cellular RNA was reverse-transcribed and subjected to RT-PCR and RT-qPCR. The primer sequences are shown in Table 4.

## Figures and Tables

**Figure 1 ijms-25-01239-f001:**
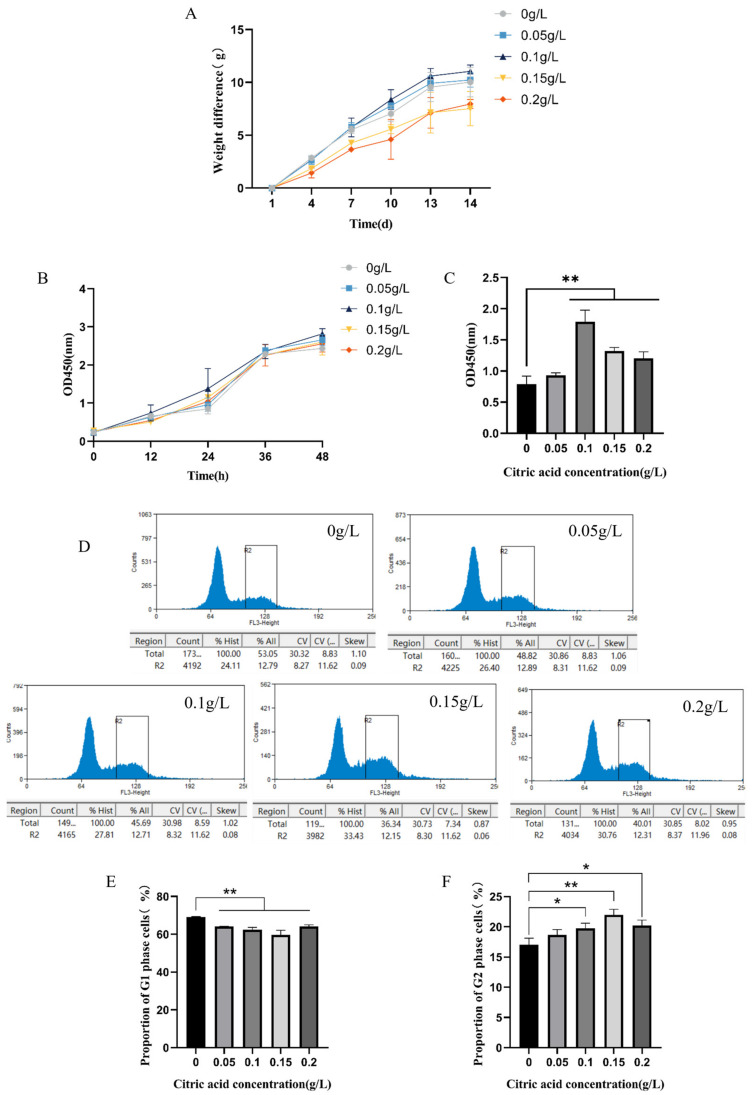
CA promotes body weight and affects IEC-6 cell growth and cycle in mice. (**A**): Mouse body weight growth curve at 14 days; (**B**): IEC-6 cell growth curve at 48 h. (OD value on the vertical axis); (**C**): Significant changes in cells after 24 h at each cell concentration; (**D**): Analysis of G1 and G2 phases in IEC-6 cells using Beckman Coulter Ultra-High-Speed Flow Sorting System; (**E**): Relative expression of cell numbers in G1 phase; (**F**): Relative expression of cell numbers in G2 phase. * Significant difference (*p* < 0.05), ** highly significant difference (*p* < 0.01).

**Figure 2 ijms-25-01239-f002:**
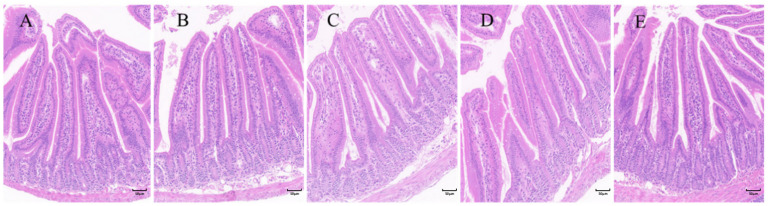
HE-stained image of small intestine (200×). (**A**): 0 g/L; (**B**): 0.05 g/L; (**C**): 0.1 g/L; (**D**): 0.15 g/L; (**E**): 0.2 g/L. Scale bar = 50 μm.

**Figure 3 ijms-25-01239-f003:**
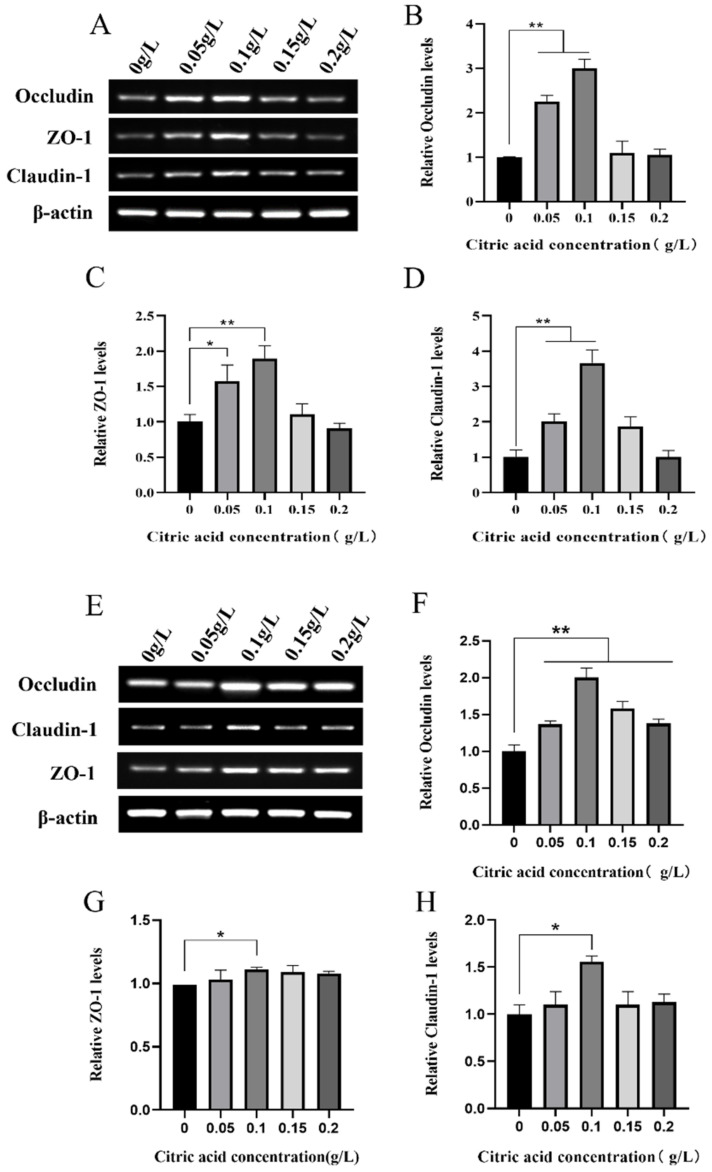
Citric acid promotes expression of intestinal tight junction genes in mice. (**A**) RT-PCR for detection of relative mRNA expression of mouse intestinal tight junction genes occludin, ZO-1, and claudin-1; (**B**–**D**) RT-qPCR for detection of relative mRNA expression of mouse intestinal tight junction genes occludin, ZO-1, and claudin-1; (**E**) RT-PCR to detect the relative expression levels of mRNA of the tight junction genes occludin, ZO-1, and claudin-1 in IEC-6 cells. (**F**–**H**) RT-qPCR to detect the relative expression levels of mRNA of the tight junction genes occludin, ZO-1, and claudin-1 in IEC-6 cells. * Significant difference (*p* < 0.05), ** highly significant difference (*p* < 0.01).

**Figure 4 ijms-25-01239-f004:**
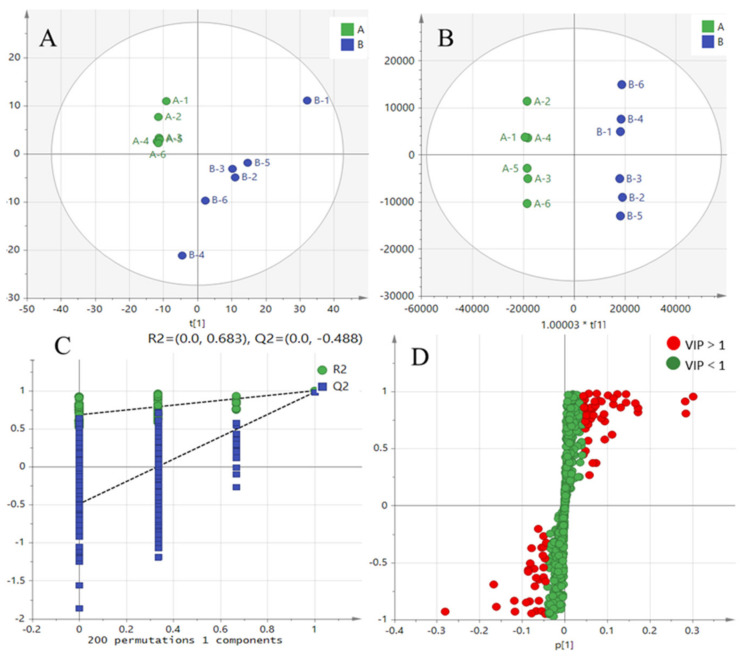
Multivariate statistical analysis. (**A**) Peca score, (**B**) PLS-DA score, (**C**) 200 displacement tests, and (**D**) S-plot test.

**Figure 5 ijms-25-01239-f005:**
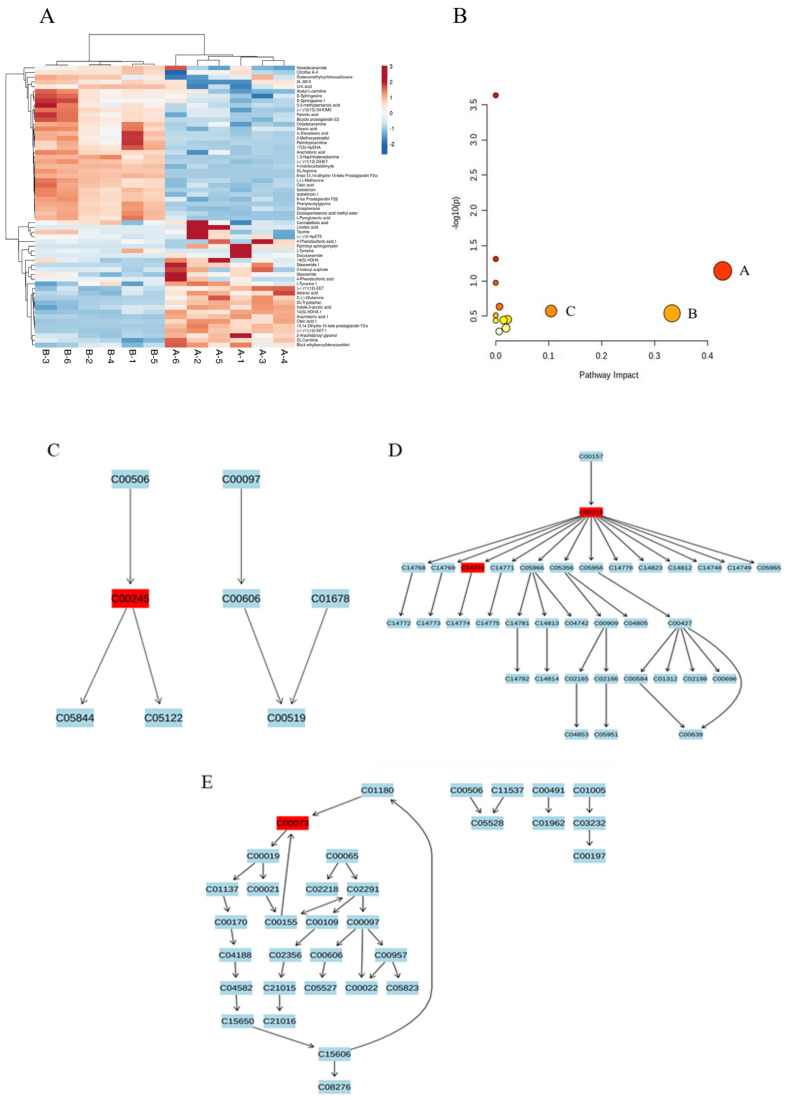
Analysis of significantly different metabolites in serum. (**A**) Heat maps of significantly different metabolite clusters, (**B**) significantly up-regulated metabolite enrichment maps, (A: linoleic acid metabolism B: taurine and subtaurine metabolism; C: arachidonic acid metabolism; the abscissa represents the pathway impact value calculated by path topology analysis, the ordinate −log10 (P) represents the significance level of metabolic pathway enrichment analysis, and the circle size represents the ratio of pathway impact to −log10 (P). The larger the ratio, the larger the circle diameter.). (**C**) KEGG pathways for taurine and hypotaurine, (**D**) KEGG pathways for arachidonic acid, and (**E**) KEGG pathways for cysteine and methionine metabolism.

**Figure 6 ijms-25-01239-f006:**
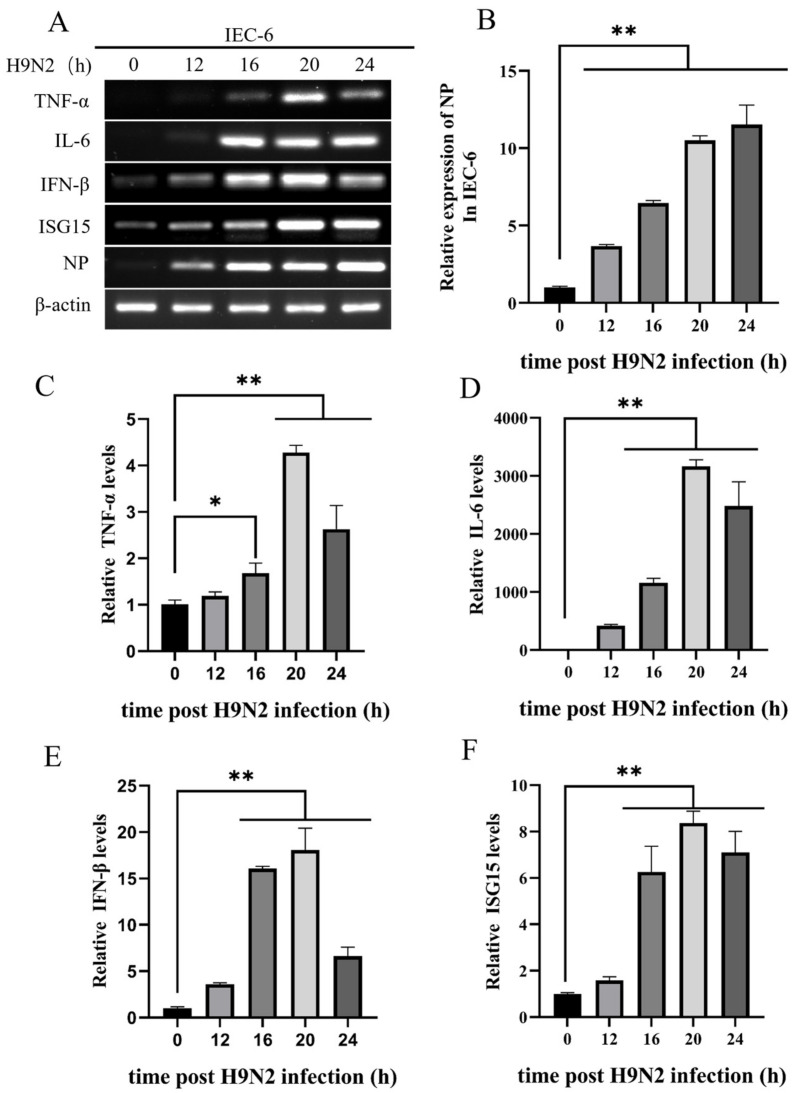
H9N2 AIV infection induces mRNA expression levels in the IEC-6 inflammatory response. (**A**) Relative mRNA expression levels of TNF-α, IL-6, IFN-β, ISG15, and viral NP detected by RT-PCR; (**B**–**F**) Relative mRNA expression levels of TNF-α, IL-6, IFN-β, ISG15, and viral NP detected by RT-qPCR. * Significant difference (*p* < 0.05), ** highly significant difference (*p* < 0.01).

**Figure 7 ijms-25-01239-f007:**
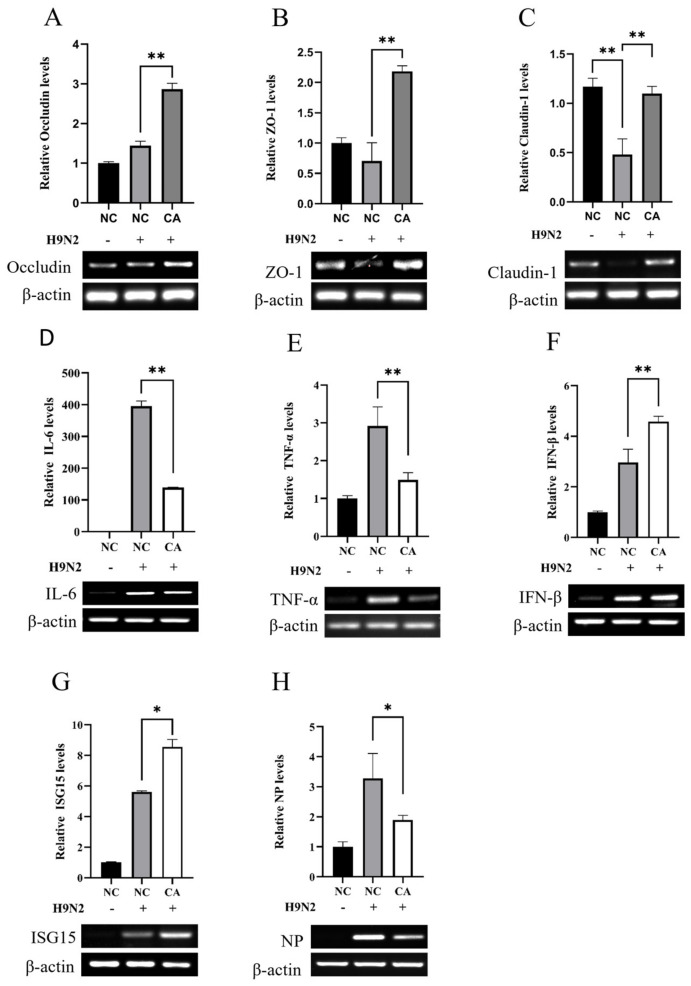
CA promotes expression of tight junction genes in IAV infection to attenuate inflammatory response and inhibit viral replication. (**A**) Relative mRNA expression levels of occludin detected by RT-qPCR and RT-PCR; (**B**) Relative mRNA expression levels of ZO-1 detected by RT-qPCR and RT-PCR; (**C**) Relative mRNA expression levels of claudin-1 detected by RT-qPCR and RT-PCR; (**D**) RT-qPCR and RT-PCR to detect the relative mRNA expression level of IL-6; (**E**) RT-qPCR and RT-PCR to detect the relative mRNA expression level of TNF-α; (**F**) RT-qPCR and RT-PCR to detect the relative mRNA expression level of IFN-β; (**G**) RT-qPCR and RT-PCR to detect the relative mRNA expression level of ISG15; and (**H**) RT-qPCR and RT-PCR to detect the relative mRNA expression level of NP. * Significant difference (*p* < 0.05), ** highly significant difference (*p* < 0.01).

**Table 1 ijms-25-01239-t001:** Comparison of the morphology and structure of small intestinal mucosa.

Groups (g/L)	Villus Height to Crypt Depth Ratio
0	3.944 ± 0.2297
0.05	4.373 ± 0.2492
0.1	5.011 ± 0.2797 ^AA^
0.15	4.720 ± 0.1643 ^AA^
0.2	4.095 ± 0.6054

Note. Lack of annotation on the data points indicates a non-significant difference. Data points showing the same uppercase letters on the shoulder labels indicate a significant difference compared with the control group (*p* < 0.01).

**Table 2 ijms-25-01239-t002:** Effect of citric acid on the community flora of the contents of the cecum of the mouse.

Bacterial Type	0 g/L	0.05 g/L	0.1 g/L	0.15 g/L	0.2 g/L
*Bifid.*	2.15 ± 0.03	2.09 ± 0.02	2.52 ± 0.01 ^AA^	2.34 ± 0.02 ^aa^	2.10 ± 0.07
*Lac.*	2.38 ± 0.03	2.37 ± 0.02	2.50 ± 0.08	2.45 ± 0.06	2.47 ± 0.15
*E. coli*	2.72 ± 0.01	2.72 ± 0.01	2.73 ± 0.01	2.71 ± 0.01	2.72 ± 0.01

Note. Lack of annotation on the data points indicates a non-significant difference. Data points sharing the same lowercase letters on the shoulder labels indicate a significant difference (*p* < 0.05). Data points showing the same uppercase letters on the shoulder labels indicate a significant difference compared with the control group (*p* < 0.01).

**Table 3 ijms-25-01239-t003:** Analysis of the pathways by which citric acid affects serum MetPA metabolism in mice.

Name	Impact	Details
Taurine and hypotaurine metabolism	0.42857	KEGG
Arachidonic acid metabolism	0.33292	KEGG
Cysteine and methionine metabolism	0.10446	KEGG
Primary bile acid biosynthesis	0.02239	KEGG
Purine metabolism	0.01945	KEGG
Fatty acid biosynthesis	0.01473	KEGG
Glutathione metabolism	0.00709	KEGG
Steroid hormone biosynthesis	0.00653	KEGG
Biosynthesis of unsaturated fatty acids	0	KEGG
Phenylalanine metabolism	0	KEGG
Fatty acid elongation	0	KEGG
Aminoacyl-tRNA biosynthesis	0	KEGG

Note. Impact: the pathway impact value calculated from pathway topology analysis. Details: the metabolic pathway database source.

**Table 4 ijms-25-01239-t004:** Primer sequences.

Gene	GenBank No.	Direction	Sense and Antisense Primer (5′–3′)
*Occludin*	NM_029865.2	Forward	GTACCCACCAGTGACCAACA
		Reverse	GTTGCTGGAGCTTAGCCTGT
*ZO-1*	XM_036152895.1	Forward	CGAGGCATCATCCCAAATAAGAAC
		Reverse	TCCAGAAGTCTGCCCGATCAC
*Claudin-1*	NM_016674.4	Forward	TGGTAATTGGCATCCTGCTG
		Reverse	CAGCCATCCACATCTTCTGC
*Bifid.*	AB064848.1	Forward	GATTCTGGCTCAGGATGAACGC
		Reverse	CTGATAGGACGAGACCCAT
*Lac.*	OM349561.1	Forward	AGCAGTAGGGATCTTCCA
		Reverse	ATTTCACCGCTACACATG
*E.coli*	OU548744.1	Forward	GGGAGTAAAGTTAATACCTTTGCTC
		Reverse	TTCCCGAAGGCACATTCT
*TNF-α*	XM_021218152.1	Forward	GAGGCACTCCCCCAAAAGAT
		Reverse	GAGGGAGGCCATTTGGGAAC
*IL-6*	NM_031168.2	Forward	GTCCTTCCTACCCCAATTTCCA
		Reverse	TAACGCACTAGGTTTGCCGA
*IFN-β*	NM_010510.2	Forward	AACTCCACCAGCAGACAGTG
		Reverse	GGTACCTTTGCACCCTCCAG
*ISG15*	NM_015783.3	Forward	CGGGAACAAGTCCACGAAG
		Reverse	CCCTCAGGCGCAAATGCT

## Data Availability

The dataset generated and analyzed in this study can be obtained from the corresponding authors upon reasonable request.

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
