# Peer review of "Citric Acid Promotes Immune Function by Modulating the Intestinal Barrier"

_ijms, 2024, doi:10.3390/ijms25021239_

Round 1

Reviewer 1 Report

Comments and Suggestions for Authors

This paper explores the use of citric acid (CA) as a preventive measure instead of antibiotics to protect intestines from infection-related damage. They have shown that CA maintains tight junctions in the intestine and hence prevents the organism from infection by decreasing permeability to the pathogen. CA increases the ratio of villi-crypts and this maintains the structural and functional integrity of the intestine. In addition, CA also reduces the inflammatory response caused by virus infection. The findings in this paper are potentially very useful to the scientific community, However, the authors need to address some minor points before it is suitable for publication.

-          Supplementary files could not be opened with any available software. The authors should use PDF or Word files, so they will be easily accessible.

-          They have mentioned that CA improves tight junctions upon infection. Have they tested leakiness of the gut before and after infection and before and after CA treatment? Most of the data involves RNA expression of tight junction-related genes but it would be better if they include some functional data also. Tight junction staining would also have been useful. HE staining in Figure 2 does not show any effect on tight junctions.

-          The authors have often mentioned figure numbers with panel alphabets throughout the paper. It gives much clarity if they write it as Figure 1A instead of just Figure A and do not let the reader guess the number based on where it appears in the paper.

-          Figure 1C: text is not legible in the figure. The authors should have used a clear image for the figure.

-          The authors should have moved current Figure 1D to the place of 1C as it flows better that way. It is very confusing for a reader to move from 1B to 1D and then back to 1C.

-          Figure 2 does not have a scale bar. There is some branching of villi with increasing doses. The authors should comment on that.

-          Table 1 legends indicate “Data points sharing the same lowercase letters on the shoulder labels indicate the significant difference”. However, there are no lowercase letters in the table. Does that mean that there is either no significant difference or a significant difference of 0.01?

-          Figure 5B: what do A, B, and C indicate? It is neither clear from the graph nor the figure legends.

-          Figure 6: what might be the reason for a decrease in inflammatory response genes at 24 hours post H9N2 infection? The authors should comment on that.

-          Line 354: what is COVID-11?

-          What gender mice were used for the experiments? The authors have mentioned that same-size mice were used but they did not mention the gender. Male and female intestines are very different in terms of size and cell numbers.

-          Line 438: it states “ Detection of the treated cells was executed/analyzed? “ what does it mean?

-          Line 457: Proper formatting is required for the formula mentioned for calculating copy number.

-          Why there is no difference in ZO-1 in cell lines in RT-qPCR in Figure 3G, whereas it is showing an increase in concentration from 0.1g/L onwards in Figure 3E RT-PCR?

-          The authors should do proper proofreading and formatting of the paper. e.g. in vivo should be in italics in lines 38, 337, etc., and there is unequal spacing throughout the paper. 

Comments on the Quality of English Language

The overall use of English is satisfactory, However, they need to do proper proofreading for the paper. 

Author Response

Dear reviewer,

Thank you for your comments on our manuscript titled 'Citric Acid Promotes Immune Function by Modulating the Intestinal Barrier' (ID: ijms-2793214). We appreciate the valuable feedback and have carefully considered the comments to improve the paper.Please see the attachment.

Reviewer 2 Report

Comments and Suggestions for Authors

Peng-Cheng Hu et al., Presented research on Citric acid promotes immune function....intestinal barrier, which is very interesting topic.

Authors presented results and conclusions are very clearly however, there are few points to addressed by authors.

1. I didn't see any control group in you results figures and table, please check and modify the figure and methodology.

2. What was the reason and why 0.1g/L CA increased Bifidobacteria rather 0.2g/L was decreased? how could we conclude CA impacting in intestinal barrier though there was not differences was seen in other bacteria too.

3. Authors are advised to do few inflammatory protein markers  which give strong evidence to the paper.

4. It is better if authors shorten the discussion.

Author Response

(The authors gave the same response as above.)
